# Genetic Transformation of *Quercus ilex* Somatic Embryos with a Gnk2-like Protein That Reveals a Putative Anti-Oomycete Action

**DOI:** 10.3390/plants11030304

**Published:** 2022-01-24

**Authors:** Susana Serrazina, Mª Teresa Martínez, Vanesa Cano, Rui Malhó, Rita Lourenço Costa, Elena Corredoira

**Affiliations:** 1Faculdade de Ciências, BioISI—Biosystems & Integrative Sciences Institute, Universidade de Lisboa, 1749-016 Lisbon, Portugal; smserrazina@fc.ul.pt (S.S.); r.malho@fc.ul.pt (R.M.); 2Misión Biológica de Galicia, Consejo Superior de Investigaciones Científicas (MBG-CSIC), Avda Vigo s/n, Campus Vida, Apartado 122, 15705 Santiago de Compostela, Spain; temar@iiag.csic.es (M.T.M.); V.CanoLazaro@enzazaden.nl (V.C.); 3Instituto Nacional de Investigação Agrária e Veterinária, Avenida da República, Quinta do Marquês, 2780-159 Oeiras, Portugal; rita.lcosta@iniav.pt; 4Centro de Estudos Florestais, Instituto Superior de Agronomia, Universidade de Lisboa, Tapada da Ajuda, 1349-017 Lisbon, Portugal

**Keywords:** antifungal protein, dehesa, *Ginkbilobin-2 gene*, in vitro tolerance assay, Holm oak, *La seca*, oak decline, *Phytophthora cinnamomi*

## Abstract

Holm oak is a key tree species in Mediterranean ecosystems, whose populations have been increasingly threatened by oak decline syndrome, a disease caused by the combined action of *Phytophthora cinnamomi* and abiotic stresses. The aim of the present study was to produce holm oak plants that overexpress the Ginkbilobin-2 homologous domain gene (*Cast_Gnk2-like*) that it is known to possess antifungal properties. Proembryogenic masses (PEMs) isolated from four embryogenic lines (Q8, E2, Q10-16 and E00) were used as target explants. PEMs were co-cultured for 5 days with *Agrobacterium* EHA105pGnk2 and then cultured on selective medium containing kanamycin (kan) and carbenicillin. After 14 weeks on selective medium, the transformation events were observed in somatic embryos of lines Q8 and E2 and a total of 4 transgenic lines were achieved. The presence of the *Cast_Gnk2-like* gene on transgenic embryos was verified by PCR, and the number of transgene copies and gene expression was estimated by qPCR. Transgenic plants were obtained from all transgenic lines after cold storage of the somatic embryos for 2 months and subsequent transfer to germination medium. In an in vitro tolerance assay with the pathogen *P. cinnamomi*, we observed that transgenic plants were able to survive longer than wild type.

## 1. Introduction

In recent decades, global climate change has greatly exacerbated the effects of many plant diseases, especially fungal diseases, which can severely affect forest ecosystems [1,2]. One of the ecosystems thus affected is the Spanish dehesas and Portuguese montados (dehesas, hereafter), which are experiencing an unprecedented crisis, fundamentally caused by a complex disease commonly known as oak decline [3]. This syndrome has drastically affected the sustainability of the dehesa ecosystem, which is unique within Europe, and the situation is of great concern due to the associated high economic, ecological and social losses [4]. Oak decline consists of the death of thousands of cork oaks and holm oaks, which are keystone species in the dehesas. Holm oak (*Quercus ilex* L.) is the most characteristic tree species in Mediterranean forests, with the largest areas of cover located in the Iberian Peninsula [5]. This species contributes to the direct production in dehesas, providing acorns, wood, comestible fungi, pollen and other products consumed by humans and animals [6]. The oomycete *Phytophthora cinnamomi* Rands is considered to be the main cause of oak decline, but the influence of other biotic and abiotic factors related to climate change accelerates the symptoms [7]. *P. cinnamomi* is one of the most destructive and widely spread oomycetes throughout the world, and it has been associated with the death of numerous different plant species, including many woody species [8]. To date, there are no efficient chemical methods available for controlling this oomycete [9]. The most effective means appears to be the afforestation of affected zones with genotypes that are tolerant to *P. cinnamomi*. However, traditional improvement programmes involving selective crossing to produce trees that are resistant/tolerant to the disease have not been carried out. Such programmes will probably never be carried out as the species has a long reproductive cycle, and thus long periods are required to complete one cycle of conventional improvement [10]. Conventional breeding methods must be complemented with biotechnological tools to produce superior trees with enhanced disease tolerance within a shorter time. In this respect, biotechnological tools such as somatic embryogenesis and genetic transformation provide a tremendous opportunity to improve tree characters [11,12,13,14]. Somatic embryogenesis is considered the best regeneration method for producing transgenic plants in hardwood species as the regeneration capacity is higher and the incidence of chimera is lower than in other regeneration methods [15]. Recent progress in *Agrobacterium tumefaciens*-mediated genetic transformation of somatic embryos of different species in the *Fagaceae* family, including holm oak, have provided the opportunity to produce transgenic genotypes [16,17,18,19]. These studies have investigated overexpression of the pathogenesis-related thaumatin-like protein in order to induce tolerance to *P. cinnamomi*. This indirect approach, i.e., the transformation with unspecific genes, was applied as it was not known which genes specifically control tolerance to the most important pathogens of these tree species. However, great advances have been made in genomic tools in recent years, and it is now possible to identify candidate genes that are directly involved in resistance in these species [20]. For example, to elucidate chestnut defense mechanisms to ink disease, root transcriptomes of the susceptible species *Castanea sativa* and the resistant species *C. crenata* were compared after *P. cinnamomi* inoculation, and a set of candidate genes of resistance was selected [21]. The expression of eight of them was quantified by digital PCR in *Castanea* genotypes showing different susceptibility to the pathogen after phenotyping through root inoculation with *P. cinnamomi* [22]. Seven of the eight candidate genes displayed differentially expressed levels depending on genotype and time-point after inoculation. *Cast_Gnk2-like* revealed to be the most expressed gene across all experiments and the one that best discriminates between susceptible and resistant genotypes [23]. The highest *Cast_Gnk2-like* expression registered in devoid of oomycete conditions also suggests that *C. crenata* root surrounding may be a hostile environment for fungal and fungal-like pathogens, such as *P. cinnamomi.*

Ginkbilobin-2 (Gnk2) is a seed storage protein present in gymnosperm *Ginkgo biloba* seeds that possesses an antifungal activity. Gnk2 has a plant-specific cysteine-rich motif DUF26 (domain of unknown function 26, also known as stress-antifungal domain: PF01657) which belongs to cysteine-rich receptor-like kinases (CRKs) not showing any similarity with other known antimicrobial proteins. Therefore, *Cast_Gnk2-like* gene may prevent pathogen growth either by its chemical properties or by inducing HR-related cell death. Miyakawa et al. [24] report that Gnk2 functions as a lectin and hypothesize that its carbohydrate-binding properties are tightly related to its fungal activity. It binds with high affinity to D-mannose and with less affinity to D-glucose, and both exist in the hyphal cell walls of *Phytophthora* species [25]. Cox et al. [26] raised the possibility that binding of lectins to mannose residues of the *Phytophthora* cell wall may confer disturbance and disruption of the cell wall structure.

The aim of the present paper was to investigate if the overexpression of the *C. crenata Ginkbilobin-2* homologous domain gene (*Cast_**Gnk2-like*) enhances tolerance to *P. cinnamomi*. To this purpose, embryogenic cultures derived from adult trees of holm oak were transformed with the *Cast_Gnk2-like* gene and the disease response against this pathogen of the regenerated transgenic plants was evaluated.

## 2. Results

### 2.1. Genetic Transformation of Proembryogenic Masses

Proembryogenic masses of four embryogenic lines of holm oak, named Q8, Q10-16, E00 and E2, were transformed with the EHA105pK7WG2-Gnk2 strain. After 10 weeks on the selection medium, the percentage of kanamycin (kan)-resistant explants was recorded. Genotype significantly affected this parameter (*p* ≤ 0.05), and new somatic embryos and/or nodular structures were only observed on lines Q8 and E2, with Q8 being the highest percentage (5%) (Table 1; Figure 1a). The resistant embryos in both lines appeared after 8 weeks of culture in selection medium, and in general, the target explants presented better appearance and lower level of necrosis in these lines than the initial explants of lines E00 and Q10-16. Negative controls (explants not transformed but cultured in selection medium) necrotized and died without forming callus or somatic embryos. Kan-resistant explants were isolated and transferred to selection medium but supplemented with 125 mg/L kan rather than 100 mg/L initially. After four weeks of cultivation under these conditions (14 weeks in total), the efficiency of transformation based on the embryo fluorescence of kan-resistant explants (Figure 1b,c) was calculated, and values of 2.5% were obtained in both genotypes (Table 1). The other two genotypes (Q10-16 and E00) did not render transformed explants. Selection efficiency ranged from 50 to 75%, and the highest figure for selection efficiency (75%) was found in genotype E2 (Table 1).

### 2.2. Maintenance of Transgenic Embryogenic Lines

To establish putative transgenic lines, an embryo was isolated at the cotyledonary stage that showed distinct GFP expression uniformly distributed over the entire embryo surface, discarding those that were not totally fluorescent. The selected embryos were proliferated in selection medium. In total, four transgenic lines were obtained, three lines from the Q8 line and one line from the E2, which were maintained by secondary embryogenesis. Once the transgenic lines were established, they were also cryopreserved according to [18] for indefinite conservation while their transformation and tolerance were tested.

### 2.3. Molecular Analysis of Transgenic Embryogenic Lines

Once the transgenic lines were established, different molecular techniques were applied to confirm the presence of *Cast_Gnk2-like* gene, the gene copy number and the transcription and expression of the gene in the transgenic lines of holm oak.

#### 2.3.1. Gene Presence Analysis

The presence of the transgenes *NPTII* and *GFP* was confirmed in all transgenic lines analyzed and in the plasmid (positive control) but not in the non-transgenic lines (WT) (negative control) (Figure 2a,b). For confirming the presence of the *Cast_**Gnk2-like* gene, two fragments of different lengths were amplified, the shorter (890 bp) including part of the T-35S terminator and part of the *Cast_**Gnk2-like* sequence, whereas the larger (1227 bp) included part of the CaMV35S promoter and the sequence of the *Cast_**Gnk2-like* gene (see Appendix A). The DNA of putatively transgenic lines produced both fragments after the amplification by PCR, confirming the transgenic nature of these lines (Figure 2c,d). Both fragments were not amplified in non-transformed counterparts (Figure 2c,d).

#### 2.3.2. Gene Number Copy Analysis

The *Cast_Gnk2-like* copy number in each transformed line was estimated through its promoter CaMV35S, by qPCR, in the lines formerly analyzed by PCR. The correlation efficiencies of the standard curve obtained as in [27] and described in the Material and Methods were between 0.993 and 0.996, and the efficiency of the reactions was between 94 and 97%. In the transformed lines, the correlation between the C_T_ and the quantity of plasmid that mimicked different copy numbers of the CaMV35S promoter pointed to one copy in lines Q8-Gin1 and E2-Gin1, two copies in Q8-Gin2 and, as expected, no copies in non-transformed genotypes (wild-type WT, Table 2). The presence of no more than two copies of the transgene may avoid silencing events the in the transformed lines.

The WT genotypes resulted in a detected C_T_, however, it was remarkably higher than the C_T_ obtained for the one copy transformed lines. On other hand the Tm of the WT amplicon (84 °C) was distinct from the Tm of the transformed lines amplicon (80 °C) (Appendix A). As *Q. ilex* do not bear the viral CaMV35S promoter, the detected amplicon in WT points to an unspecific amplification.

#### 2.3.3. Gene Expression Analysis

*Cast_Gnk2-like* gene expression in each transformed line was evaluated in somatic embryos at early cotyledonary stage by qPCR (Figure 3). As *Q. ilex* and *C. crenata* are taxonomically close, the endogenous *Gnk2-like* gene in *Q. ilex* may have a high level of similarity with *Cast_Gnk2-like*, and both genes can be amplified in the same reaction. However, the expression level obtained for the two *Q. ilex* non-transformed genotypes was lower than the expression level obtained for the respective transformed lines. This was expected, as the transgene *Cast_Gnk2-like* is guided by the constitutive promoter CaMV35S.

In the transformed lines, Q8-Gin1, Q8-Gin3 and E2-Gin1 had significantly (*p* ≤ 0.05) higher expression levels of *Cast_Gnk2-like* when compared to the non-transformed genotypes, indicating an overexpression of the *Cast_Gnk2-like* gene. Q8-Gin1 is the line with the highest transgene expression. Despite the estimation of two copies of the transgene in Q8-Gin2, double of what was found in the other transgenic lines, it was not reflected in a significantly higher expression of the *Cast_Gnk2-like* gene. On the contrary, the expression in Q8-Gin2 was the lowest amongst the transgenic lines.

### 2.4. Plant Regeneration from Transgenic Somatic Embryos

After cold storage and germination treatments, root-only development and whole plant recovery (Figure 1d) were obtained in somatic embryos of four holm oak transgenic lines and in their corresponding wild-type lines (Table 3). Among lines derived from Q8 lines, conversion rates ranged from 30.6% to 69.5% and without significant differences to one another (Table 3). Of the three transgenic lines, only in the Q8-Gin3 line were the conversion percentages lower than those registered in the non-transformed line. In line E2, the conversion frequencies were higher in the non-transformed line (38.9%) than in E2-Gin1 (22.2%), but no significant differences were detected between transgenic and wild type material (Table 3). In addition, not significant differences were found in the shoot length and leaf number.

Leaves and roots of the regenerated plantlets were evaluated regarding GFP expression. Fluorescence was detected in shoots and roots of transgenic plantlets (Figure 1e,f), indicating stable transformation of these lines. No GFP fluorescence was observed in non-transformed shoots and roots subjected to the same excitation and emission conditions than transgenic material. Plants transformed with *Cast_Gnk2-like* gene grew normally and were morphologically indistinguishable from non-transformed controls (Appendix A).

### 2.5. In Vitro Tolerance Assay

Transformed plants of holm oak and its untransformed counterparts obtained as described above were infected in order to determine if the overexpression of *Cast_Gnk2-like* gene could improve the tolerance against *P. cinnamomi*. To test this, the number of survived days of transgenic and non-transformed lines was evaluated after plant inoculation with the oomycete. Infection signs were more evident, and its progression was faster in non-transgenic plants than in transgenic plants, which appear completely fulminated between 4 and 6 days (Figure 4). All transgenic lines derived from genotype Q8 survived longer than the untransformed control line, but only the Q8-Gin1 line (10.35 days) showed significantly (*p* < 0.05) higher survival ability than the control line (4.46 days) (Figure 5). Similarly, line E2-Gin1 survived significantly more days (8.1 days) than the E2-WT line (6.69 days) (Figure 4). These results seem to be correlated with the *Cast_Gnk2-like* expression data since the Q8-Gin1 and E2-Gin2 lines showed the highest levels of expression (Figure 3). Finally, all plants survived in control conditions (plants grown in the same in vitro tolerance assay medium but without the oomycete), which indicated that culture in liquid medium does not have a detrimental effect on the plants.

## 3. Discussion

Oak decline syndrome has been recognized as one of the most important ecological problems in Europe in the last few decades [5,28]. Among oak species, holm oak is considered one of the most susceptible to the syndrome. Therefore, improving holm oak tolerance to the disease has emerged as a subject of great research interest, particularly with respect to transformation studies. In the present study, we investigated whether overexpression of the *Cast_Gnk2-like* gene from *C. crenata* in somatic embryos of holm oak increases the tolerance of the trees to *P. cinnamomi*, the main cause of oak decline. This gene codifies a seed storage protein present in the gymnosperm of *Ginkgo biloba* seeds. Thisprotein, named ginkbilobin has exhibited potent antifungal activity against diverse fungi, including Botrytis cinerea, Mycosphaerella arachidicola, Fusarium oxysporum, Rhizoctonia solani and Coprinus comatus [29,30]. *Ginkbilobin* proteins may prevent the growth of pathogens by chemical reactions or by inducing hypersensitive response-related cell death. Several antifungal proteins such as defensins, lipid tranfer proteins and thionins have been successfully overexpresed in order to confer tolerance to diseases caused by different fungi [31,32,33]. However, and despite the obvious antifungal properties of ginkbilobin, this protein has scarcely been studied, and there are very few reports of genetic transformation with this protein. Cucumber transgenic plants overproducing the *Gnk2-1* gene have displayed resistance to *Fusarium oxysporum* [34]. McGuigan et al. [35] were recently successful in overexpressing the *Cast_Gnk2-like* gene in American chestnut somatic embryos, but the tolerance of transgenic plants to *P. cinnamomi* has not yet been checked.

Although the number of transformation systems has increased greatly in recent years, production of transgenic plants remains difficult and is limited to a small number of species and genotypes [36,37]. This is particularly true in the case of woody species, many of which are recalcitrant to transformation [36] and references therein]. Holm oak belongs to the group of woody species that are considered recalcitrant to regeneration and transformation [10]. However, the present study reports the successful transformation of holm oak somatic embryos with the *Cast_Gnk2-like* gene. A key factor in achieving genetic transformation in this species is the use of PEMs as target explants. This type of material has already been suggested to be the most suitable target tissue for the transformation of several woody species such as *Vitis rotundifolia* [38], *C. dentata* [39] and *Coffea arabica* [36]. As in other transformation systems [19,39,40], transformation efficiency in holm oak is affected by the genotype of the embryogenic lines, and transformation was only obtained in two of the four lines evaluated. Although the transformation efficiency was low (2.5%), it was similar to the values reported in a previous paper (1.66–3.33%) on holm oak genetic transformation [18]. Low transformation rates have also been reported for other recalcitrant species, cultivars and genotypes. For example, transformation efficiency of less than 1% was observed in the recalcitrant species *C. arabica* [41]. Low transformation rates (0.3–4%) were also obtained in a recalcitrant pear cultivar [42]. Likewise, for avocado, another species considered recalcitrant to transformation, a transformation rate of only 10% has been reported [43].

The selection efficiency was relatively high in the present study, especially in line E2. The results validate GFP fluorescence as an effective, precise and non-destructive marker for detecting transgenic somatic embryos in holm oak. Moreover, a strong GFP fluorescence signal was detected in holm oak somatic embryos and in different tissues derived from transgenic plants. Double selection via antibiotic resistance and expression of a GFP gene has also been successfully applied in other recalcitrant species, improving the selection process and reducing the escape number, selection time and antibiotic concentration [42,44].

A series of molecular studies revealed expression of the *Cast_Gnk2-like* gene and estimated the copy number. Southern blotting constitutes the most precise technique to evaluate copy number, but it requires high amounts of genomic DNA, skills and is time-consuming. Instead, qPCR was chosen as it is sensitive and permits specific quantification in a high number of samples with low DNA quantities [27,45]. As generally described in most genetic transformation systems, the level of transgene expression varied greatly among the different transgenic lines generated. This is due to the copy number as well as to other factors such as the integration site and the presence of incomplete copies [46,47]. A similar problem has also been mentioned in regard to gene expression in other woody species such as European chestnut [16], olive [48] and cork oak [19]. Low transgene copy numbers have been described in other embryogenic systems of several woody species [36,39,49]. Holm oak is no exception and only one or two copies of the Cast_Gnk2-like gene have been detected.

Although plant conversion from somatic embryos in many hardwoods is considered an important bottleneck [15], in the present study plants from four independent transgenic lines were successfully recovered by applying the protocols previously described for holm oak somatic embryos [9,50]. The in vitro behaviour and growth of these transgenic plants were not affected by overexpression of the *Cast_Gnk2-like* gene. Although the conversion frequencies were generally high, they varied considerably depending on the embryogenic line, probably because the number of gene copies, the mode of insertion and their position in the plant genome cannot be controlled [19].

An in vitro assay was used to evaluate the tolerance of transgenic plants. The advantages of in vitro tests include the fact that the rapid execution enables several repetitions to be made in a short period of time and only rooting of regenerated shoots is required. In addition, in vitro tests ensure that all plants are of the same age and are physiologically homogeneous. It can be concluded that the infection in liquid medium method is very useful for the rapid evaluation of several transformed lines, enabling initial selection of those lines showing the greatest tolerance to the pathogen. Subsequently, only transgenic lines that produce the best results can be multiplied and acclimatized, thereby reducing the time required and material used. In the present study, we found that the overexpression of the *Cast_Gnk2-like* gene conferred a moderate level of tolerance to *P. cinnamomi*. The findings confirm the significant increase in survival (days) in two lines (Q8-Gin1 and E2-Gin1), both of which have one copy of the *Cast_Gnk2-like* gene and display greater expression of the transgene than the respective wild-type genotypes. The very low copy number of the T-DNA inserted in these two lines makes them more suitable for further functional analysis, as there is a low probability of transgene inactivation [36,51].

Moreover, the level of disease tolerance was positively correlated with the level of expression of the *Cast_Gnk2-like* gene. Although ginkbilobins do not resemble other pathogenesis-related proteins, they display homology to the extracellular domain of plant cysteine-rich receptor-like kinases, which explains their role [34]. Plant cysteine-rich receptor-like kinases are known to be involved in the regulation of immune responses [52,53]. Mou et al. [54] recently reported that the overexpression of plant cysteine-rich receptor-like kinase in pepper and tobacco produces an antifungal effect against *Ralstonia solanacearum*, similar to that caused by ginkbilobin in *Ginkgo biloba*. In the present study, expression of a *C. crenata* Gnk-2 like protein was achieved in *Q. ilex*, with a putative high level of similarity with the endogenous protein and a similar function, as the two species are closely related. Thus, the tolerance to *P. cinnamomi* observed in in vitro transformed *Cast_Gnk2-like* holm oak plants suggests the involvement of the Gnk-2 like protein and indicates the need for further studies to clarify the action of the protein against oomycetes.

## 4. Materials and Methods

### 4.1. Plant Material

For the genetic transformation of holm oak, four embryogenic lines were used: E2, Q8, E00 and Q10-16. Lines E2, Q8 and E00 were induced from teguments of ovules derived from adult trees [55], while the Q10-16 line was initiated in a leaf excised from axillary shoot cultures established from a centenary holm oak [50]. Mother trees used as source of initial explants to induce embryogenic lines were growing in two locations of Spain: genotypes E in Alcala de Henares and genotypes Q in Toledo for more details see [50,55]. Despite their different origin, the 4 embryogenic lines show a similar behavior during proliferation and germination steps. Embryogenic cultures have been maintained by secondary embryogenesis with subcultures every 6 weeks, in proliferation medium that consisted of mineral medium SH [56], MS vitamins [57], 30 g/L sucrose and 6 g/L Plant Propagation Agar (PPA; Condalab, Madrid, Spain). The pH was adjusted to 5.6 before autoclaving at 115 °C for 20 min. The medium was dispensed into Petri dishes (9 cm in diameter) and 10–12 individualized proembryogenic masses were cultured on each dish.

Unless specified, the cultures were maintained in a photoperiodic climatic chamber with a 16 h light and 8 h dark photoperiod (standard conditions). In this chamber, light was supplied by tubes white light fluorescent lamps (Mazdafluor 7D TF 36 w/LJ) with a photon flux density of 50–60 μmol·m^−2^·s^−1^.

### 4.2. Binary Plasmid Construction and Transformation into Agrobacterium Strain

The chestnut *Ginkbilobin2* homologous domain gene (*Cast_**Gnk2-like*) identified by Serrazina et al. [21] in infected roots of *C. crenata* was cloned into the plasmid pK7WG2D under the CaMV35S promoter utilizing the Gateway recombination cloning technology (Thermo Fisher Scientific, Waltham, MA, USA). The vector pK7WG2D (Department of Plant Systems Biology, VIB University of Ghent, Belgium) also includes the selectable marker gene neomycin phosphotransferase (*NPTII*) and the reporter gene for green fluorescence protein (*GFP*). The cDNA sequence encoding *Cast_**Gnk2-like* was amplified by PCR applying the proofreading ADN polimerase AccuPrime™ Pfx (Thermo Fisher Scientific, Waltham, MA, USA). The PCR product was transferred into the pENTRTM/D-TOPO^®^ vector (Thermo Fisher Scientific, Waltham, MA, USA) according to the manufacturer’s instructions. To confirm the correct orientation of the *Cast_**Gnk2-like* cDNA in the pENTRTM/D-TOPO^®^ plasmid, PCR amplification was carried out and the entire insert was subsequently sequenced. The correct product was transferred into the plasmid pK7WG2D employing the Gateway™ LR Clonase™ II Enzyme Mix (Thermo Fisher Scientific, Waltham, MA, USA) as recommended the manufacturer. The binary plasmid, called pK7WG2D-Gnk2 (Appendix A), was transferred to *Agrobacterium tumefaciens* strain EHA105 [58] by the freeze-thaw method [59] and was used in the transformation procedures.

### 4.3. Transformation Procedures

The cultures of the EHA105pK7WG2-Gnk2 strain were started at from a glycerol stock stored at −80 °C and were grown overnight at 28 °C, with stirring (180–200 rpm), in a Luria–Bertani liquid (LB: 10 g/L tryptone, 0.5 g/L yeast extract and 10 g/L of NaCl, pH 7.0) [60] supplemented with 50 mg/L of kanamycin. A Petri dish with solid LB medium (1.5% agar) was inoculated with this bacterial suspension, and the dish was incubated at 28 °C in darkness. After 3 days of culture, a colony isolated from this dish was inoculated in 2 mL of liquid LB medium supplemented with 50 mg/L kan. The culture was grown overnight at 28 °C with shaking (180–200 rpm) in darkness. Next, 1 mL of this bacterial suspension was used to inoculate 600 mL of liquid LB medium with 50 mg/L kan. Again, it was incubated overnight at 28 °C and with shaking (100 rpm) in darkness until an OD600 = 0.6 was reached. Then, the bacterial culture was centrifuged at 6500 rpm for 10 min at 10 °C, and the pellet obtained was resuspended in 200 mL of liquid MS mineral medium supplemented with 5 g/L sucrose (infection medium).

For the genetic transformation, 2 or 3 individualized PEMs isolated from 6-week-old cultures after the last subculture were pre-cultured for 1 week on proliferation medium. Subsequently, explants were immersed in infection medium during 30 min with gentle shaking. After that time, the infection medium was removed by the filtration and the explants were transferred to proliferation medium for 5 days of coculture in the dark at 25 °C. The bacteria were eliminated by immersion of explants in a washing solution which consisted of sterile water with carbenicilin (300 mg/L; CB). After 30 min, the wash solution was removed by filtration, and the explants were transferred to proliferation supplemented with 300 mg/L CB and 75 mg/L kan. After two weeks of culture in standard conditions, the explants were transferred to selection medium consisting of proliferation medium supplemented with 300 mg/L CB and 100 mg/L kan. The embryos were maintained under standard conditions, with periodic subcultures every two weeks to this selection medium, for 10 weeks from the beginning of the experiment.

For each embryogenic line, 120 explants (12 Petri dishes with 10 explants) were used. Additionally, for each embryogenic line, 20 not infected explants were cultured in proliferation medium without antibiotics (control positive) and with antibiotics (negative control).

### 4.4. Selection and Proliferation of Putative Transgenic Lines

After 10 weeks of culture on selective medium, the rate of kan-resistant explants defined as the percentage of initial explants showing the formation of new somatic embryos and/or embryogenic structures was recorded. Kan-resistant embryos were isolated and transferred to medium proliferation supplemented with a higher concentration of kan than the selection medium (125 mg/L) and were cultivated for 4 more weeks in order to increase the selective pressure and confirm its resistance to kan. After 14 weeks from the start of the transformation experiments, the efficiency of transformation, defined as the percentage of initial explants that show fluorescence (GFP+) was determined. Fluorescence was observed using a Leica M205 FA (Leica, Wetzlar, Germany) stereomicroscope, equipped with a 200-W bulb, and a specific filter for fluorescence, with 470/40× nm of excitation and 525/50 nm of emission. The images were taken with a Leica DSC7000T camera (Leica, Wetzlar, Germany). Finally, selection efficiency was estimated as the percentage of kan-resistant explants that also developed GFP-positive embryos.

From each GFP + explant, a single embryo at cotyledonary stage was isolated and cultured on selective medium for at least four subcultures to establish putative transgenic embryogenic lines. During that time, GFP expression was also checked periodically to rule out the appearance of chimeras. After this time, the lines putatively transformed were transferred to proliferation medium without antibiotics, to remove selection pressure and promote growth and multiplication of the lines. These transgenic lines were maintained by secondary embryogenesis, with subcultures every 6 weeks, following the conditions previously described in the Section 4.1.

### 4.5. Molecular Confirmation of Transformation

#### 4.5.1. Gene Presence Analysis

The presence of the transgenes (*NPTII*, *GFP* and *Cast_**Gnk2-like*) was confirmed by amplifying its sequence by PCR. DNA extraction from non-transgenic and transgenic somatic embryos was carried out using the Kit REALPURE (Durviz, Valencia, Spain) following the instructions of the manufacturer and prior homogenization of the plant material (100 mg) in mortars with liquid nitrogen. DNA concentration was quantified on a Nanodrop ND-2000 spectrophotometer (Thermo Fisher Scientific, Waltham, MA, USA). All amplification reactions were carried out in an MJ Mini thermal cycler (BioRad, Hercules, CA, USA), using a reaction volume 50 μL. The reaction is composed of 250–500 ng of genomic DNA, 1 μL Cl_2_Mg, 0.2 μL Taq DNA Polymerase (Qiagen, Hilden, Germany), 2.5 mM of dNTPs and 15 µM of primers. The amplification programs used, the sequence of the primers as well as the size of the generated fragment are specified in Appendix A. The PCR amplification products were resolved on 1.5% gels agarose (Ultra Pure Agarose, Invitrogen, Waltham, MA, USA), in TBE 1X (Duchefa, Haarlem, the Netherlands) and 1% of RedSafe (iNtRON Biotechnology, Sangdaewon-Dong, Korea) dye. After the electrophoresis is complete, the gels were visualized with a transilluminator equipped with an imaging program (Quantity One 1-D Analysis Software, BioRad, Hercules, CA, USA).

#### 4.5.2. Gene Number Copy Analysis

The number of copies of the *Cast_**Gnk2-like* gene inserted in each putative transgenic line was estimated by quantitative real-time PCR (qPCR). DNA was extracted as described in the Section 4.5.1, from non-transgenic and transgenic somatic embryos and amplified independently to achieve three biological replicates for each line. As *Q. ilex* and *C. crenata* are taxonomically closely related, the nucleotide sequences of the *Gnk2-like* gene must have high similarity. Therefore, the CaMV35S promoter of viral origin that drives the expression of *Cast_**Gnk2-like* gene in transgenic lines was selected to compare the copy number of the inserted gene. Specific oligos designed for the promoter are described in Appendix A. A BLASTn was performed to the primer pair against known *Quercus* genomes in the National Center for Biotechnology Information (NCBI), with no predicted alignment to specific genes. In the present study for *Q. ilex*, there are no previous reports that reveal a reference gene or a transgenic line for which the copy number had been estimated. As in Song et al. [27], we established a standard curve by mixing the plasmid carrying the transgene used for transformation (pK7WG2D-Gnk2) with non-transformed genomic DNA. Based on the ratio of transgene copy number to the size of *Q. ilex* genome (1.98 pg/2C, [61]) we calculated the quantity of plasmid needed to be mixed with 2 ng of non-transgenic *Q. ilex* DNA to simulate 1, 2, 5 and 10 copies of transgene. In each reaction, 2 ng of DNA were used in a 20 µL final volume using 10 µL of Sensi Fast SYBR Hi-ROX kit (Bioline, Meridian Bioscience, Cincinnati, OH, EUA) and 0.2 µM of each primer in a StepOne Real-Time PCR system (Applied Biosystems, Foster City, CA, USA). qPCR conditions were the same used in [45]: 95 °C for 10 min, then 40 cycles of 95 °C for 15 s and 58 °C for 1 min. Each set of reactions included a no template control, two technical replicates and two repetitions. Upon thermal cycling, dissociation curves or melt curve analysis were used to analyze non-specific PCR products. The standard curve was set that plot the threshold cycle (C_T_) with the plasmid DNA initial quantity, and the copy number in each sample was estimated making the correspondence between the C_T_, the quantity of plasmid and the mimicked copies of transgene.

#### 4.5.3. Gene Expression Analysis

The expression of the *Cast_Gnk2-like* gene was analyzed by qPCR. Total RNA was extracted from non-transformed and putatively transformed lines using somatic embryos at early cotyledonary stage and the Kit Qiagen RNeasy Plant Mini (Qiagen, Hilden, Germany) according to the manufacturer’s instructions. Three biological replicates per line were considered. Once RNA was extracted, it was treated with the RNase-free DNase Set (Qiagen, Hilden, Germany) to eliminate a possible DNA contamination. Stability of the extracted RNA was verified by observing a double band in gels of 1.2% agarose with formaldehyde, and the concentration was quantified with a Nanodrop ND-2000 spectrophotometer (Thermo Fisher Scientific, Waltham, MA, USA). Total RNA (2 µg) was used as template for reverse transcription with RevertAid H Minus Reverse Transcriptase (Thermo Fisher Scientific, Waltham, MA, USA) and primed with an oligo(dT) primer. Specific primers for *Cast_Gnk2-like* are described in Appendix A. 9.7 ng of cDNA were used per reaction in a 15 µL final volume using 7.5 µL of Maxima SYBR Green qPCR Master Mix kit (Thermo Fisher Scientific, Waltham, MA, USA). A final concentration of 0.2 μM of each primer was used in a CFX96 Touch Real-Time PCR Detection System (BioRad, Hercules, CA, USA). Reactions started with a denaturation step at 95 °C for 10 min followed by 40 cycles of denaturation at 95 °C for 15 s and annealing temperature for 30 s. Each set of reactions included a no template control and three technical replicates. Dissociation curves were used to analyze non-specific PCR products. To normalize expression data, *Elongation factor 1 alpha* (*EF1**α*) [62] and *β-Tubulin* (*TUB*) [63] were used and the oligos, designed for species within *Quercus* genera, are described in Appendix A. Gene expression was calculated using the Pfaffl method [64]. Student’s t-test was used for statistical analysis.

### 4.6. Regeneration of Transgenic Plants

Maturation and germination of somatic embryos from non-transgenic and transgenic lines was performed according the procedure described by Martínez et al. [50]. Somatic embryos at cotyledonary stage (≥5 mm) were isolated from 6-week-old cultures and transferred to empty Petri dishes (9 cm in diameter), where they remained in semi-dark conditions for two months at 4 °C. Elapsed this time, somatic embryos were cultured in glass jars 500 mL with 70 mL of germination medium. This medium consisted of GD mineral medium [65] supplemented with 0.1 mg/L 6-benziladenine (BA) and 20 μM silver thiosulfate. For each WT and transgenic line, 36 somatic embryos were cultured per jar (6 embryos per jar). After eight weeks of culture under standard conditions, the following parameters were determined: percentage of embryos that only developed a root ≥ 5 mm and the percentage of embryos that develop a complete plant, as well as the root (mm) and shoot (mm) lengths and the number of leaves per plant. After germination, leaves and roots from transgenic plantlets and their non-transformed counterparts were also examined for stable GFP expression.

### 4.7. Plant Pathogenicity Assay

#### 4.7.1. Oomycete Strain and Sporangium and Zoospore Preparation

The *P. cinnamomi* A2 strain, named UEX-1, used in the tolerance experiments, was isolated from the infected roots of holm oak trees from Valverde de Mérida (Badajoz, Spain) [66]. The strain was kept in V8-Agar medium [67] with replicates every 2–3 weeks. Because the strain may lose its ability to infect with time, before each infection experiment, it was reactivated, using the protocol established by [68]. Once the UEX-1 strain was reactivated, it was grown for one week in disposable Petri dishes (9 cm diameter) with 25 mL V8 medium without antibiotics in darkness at 24 °C. From the outermost area from the plate, where growth was most active, discs of 5 mm agar with the help of a punch and placed on a Petri dish (9 cm diameter) with 25 mL of clarified V8 medium. After incubation for 24 h in darkness at 24 °C, the medium V8 was replaced by 25 mL of Chen and Zentmyer salt solution [69]. Subsequently, 4 washes with the salt solution every 30 min were performed. After the last wash, the mycelium discs remained in the dish, bathed with the salt solution under black light at 24 °C for 24 h. Sporulation started approximately 8 h after the last wash, and reached its level maximum between 24 and 36 h from onset. After the incubation period, the correct sporangia formation was verified with stereomicroscope and to induce synchrony sporulation and the simultaneous release of zoospores in the tube Petri dishes were incubated for 15–30 min at 4 °C just before starting the plant infection experiment.

#### 4.7.2. In Vitro Tolerance Assay

To evaluate the tolerance of transgenic holm oak plants to *P. cinnamomi*, a mycelium disc with sporangia developed as described in previous section were placed on the bottom of each glass tube with 15 mL of GD liquid medium supplemented with sucrose 3% and 0.1 mg/L BA [68,70]. Subsequently, plants derived from the germination of non-transgenic and transgenic somatic embryos were transferred to these glass tubes. The tolerance of plants against the oomycete was evaluated as the number of days of survival in an incubation period of 15 days. The plants were examined daily and a plant was considered dead when all organs were 100% necrotic. For each line, a total of six plants were infected, and the experiment was repeated thrice (i.e., 18 plants per line).

### 4.8. Data Handling and Statistical Analysis

Statistical analyses were accomplished using the SPSS 26 software (SPSS Inc., Chicago, IL, USA). Transformation results were analyzed by Chi-Square test (χ^2^). Germination and infection survival data were tested for homogeneity of variance (Levene Statistics). Thereafter, in both experiments the statistical significance was tested using analysis of variance one way (ANOVA I) and multiple mean comparisons were performed by Tukey’s HSD (honestly significant difference) test. The arcsine square-root transformation was applied to proportional data prior to analysis. Non-transformed data are presented in the Tables and Figures. For the qPCR expression analysis, Student´s t-test was applied.

## 5. Conclusions

For the first time, the overexpression of *Cast_Gnk2-like* gene which codifies a *Ginkbilobin-2* homologous domain gene reported with antifungal properties was attained. An in vitro test used to evaluate its overexpression can improve the tolerance of holm oak plants against the infection of *P. cinnamomi*. After infection, transgenic plants survived more days than non-transformed plants. Based on our data, we suggest that the *Cast_Gnk2-like* gene is a promising gene for further functional analysis studies to improve plant disease tolerance. In a near future, experiments with ex vitro plants are needed to confirm the tolerance to *P. cinnamomi,* then biochemical studies with the isolated protein will be essential to understand its action on oomycetes.

## Figures and Tables

**Figure 1 plants-11-00304-f001:**
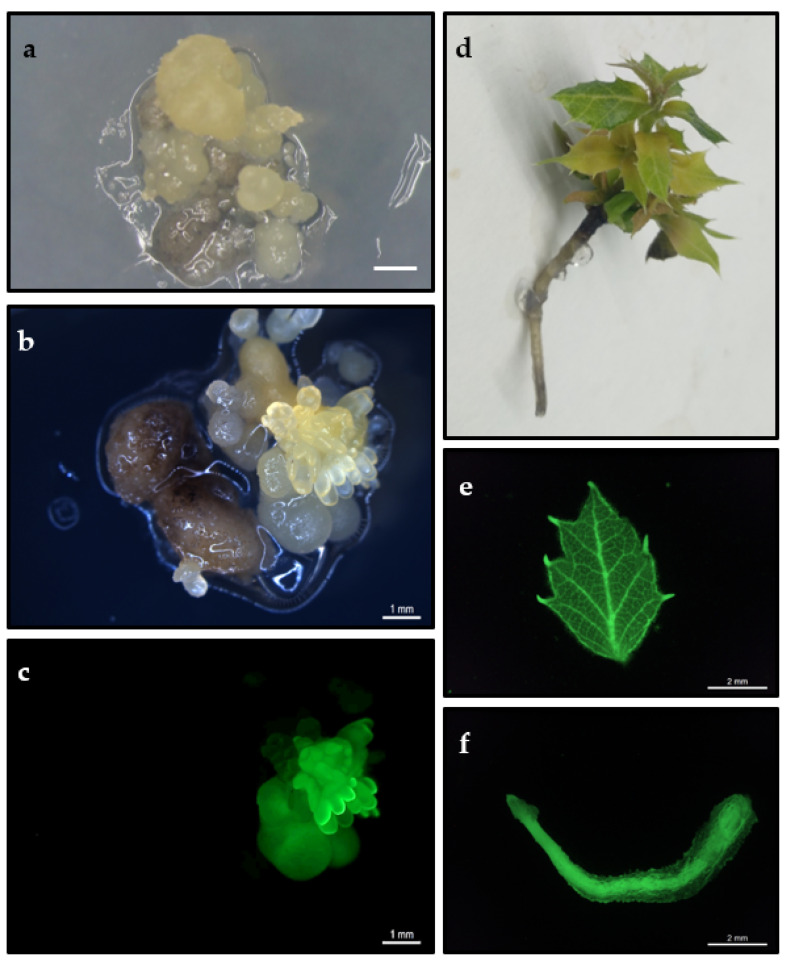
(**a**) Appearance of kan-resistant embryos developed on selection medium 10 weeks after transformation. Bar: 1 mm. (**b**,**c**) GFP expression in transformed holm oak somatic embryos observed under white light (**b**) and the same embryos observed with ultraviolet light (**c**). (**d**) Plant derived from the germination of somatic embryo from a transformed line. (**e**,**f**) GFP expression in a leaf (**e**) and in a root (**f**) excised from a transgenic plant and observed with ultraviolet light.

**Figure 2 plants-11-00304-f002:**
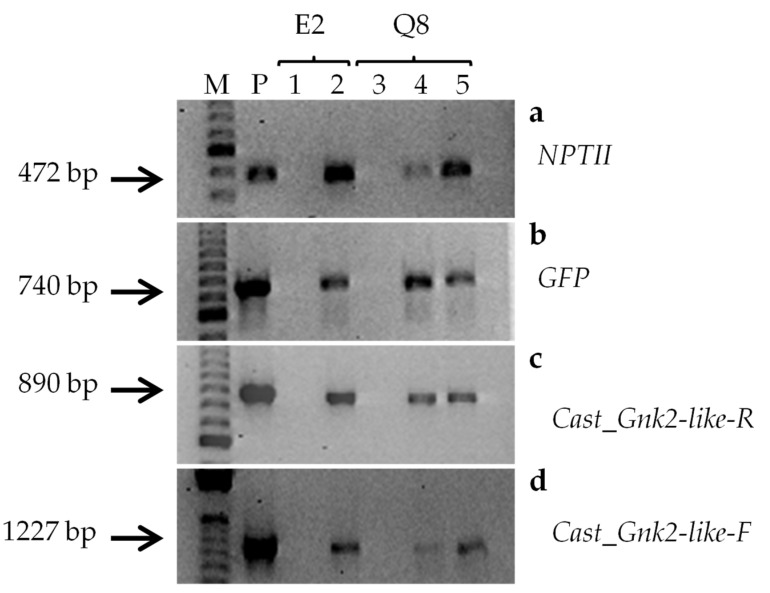
Screening of transgenic embryos by PCR. The presence of transgenes was determined by PCR screening using the primer pairs specific to (**a**) NPTII (472 bp), (**b**) GFP (740 bp), (**c**) Cast_Gnk2-like-R fragment (890 bp) and (**d**) Cast_Gnk2-like-F fragment (1227 bp). M DNA ladder; P corresponds to plasmid DNA (positive control); Lanes 1–2: E2 amplification corresponding non-transformed (1) and transformed (2) lines; Lanes 3–5: Q8 amplification corresponding non-transformed (3) and transformed (4, 5) lines.

**Figure 3 plants-11-00304-f003:**
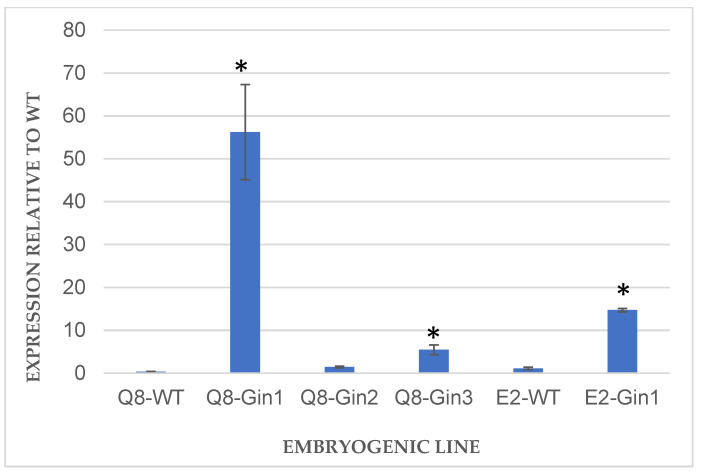
Expression analysis of *Cast_Gnk2-like* gene in somatic embryos of holm oak. Expression in transformed lines was calculated relative to the respective non-transgenic lines (WT), and normalized to *EF1*α and *TUB* reference genes. Bars are mean ± standard error (*n* = 3). Asterisks indicate significant differences in the expression when compared to WT with *p* < 0.05 (*t*-test).

**Figure 4 plants-11-00304-f004:**
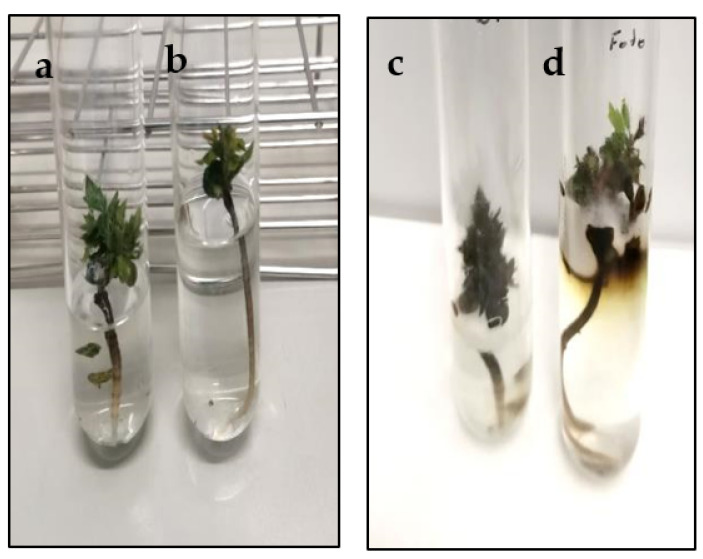
Morphological appearance and progress of infection with *P. cinnamomi* in a non-transformed plantlet (**a**,**c**) and transformed plantlet (**b**,**d**) of genotype Q8 immediately after inoculation (**a**,**b**) and 7 days later (**c**,**d**).

**Figure 5 plants-11-00304-f005:**
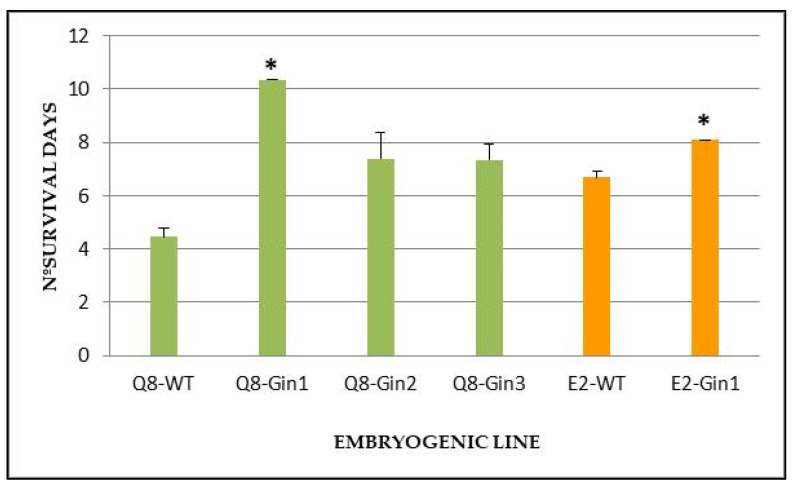
Survival ability of transformed and non-transformed (WT) holm oak plants derived from genotypes Q8 and E2 after 15 days of infection with *P. cinnamomi*. Bars are mean ± standard error (*n* = 3). Asterisks indicate significant differences in the survival when compared to WT with *p* < 0.05 (Tukey’s HSD test).

**Table 1 plants-11-00304-t001:** Effect of genotype on percentage of kanamycin-resistant explants, transformation efficiency and selection efficiency following 14 weeks of culture in selection medium.

Embryogenic Line	Kanamycin Resistant Explants (%) ^1^	Transformation Efficiency (%) ^2^	Selection Efficiency (%) ^3^
Q8	5.0 ± 2.7	2.5 ± 1.7	50.0
E2	3.3 ± 1.4	2.5 ± 1.3	75.0
Q10-16	0.0 ± 0.0	0.0 ± 0.0	-
E00	0.0 ± 0.0	0.0 ± 0.0	-
χ^2^	*p* < 0.05	ns	-

Values are means ± SE of 12 Petri dishes with 10 explants each. ^1^ Percentage of initial explants that are kanamycin-resistant after 10 weeks in selection medium. ^2^ Percentage of initial explants that are GFP+ after 14 weeks in selection medium. ^3^ Percentage of kanamycin-resistant explants which also are GFP+. ns, not significant.

**Table 2 plants-11-00304-t002:** Estimated copy number in holm oak somatic embryogenic lines transformed with *Cast_Gnk2-like* gene.

Line	C_T_ Mean	C_T_ Standard Deviation	Estimated Copy Number ^1^
Q8-WT	27.27	0.97	0
Q8-Gin1	21.55	0.61	1
Q8-Gin2	20.51	0.27	2
E2-WT	27.35	1.35	0
E2-Gin1	21.69	0.41	1

^1^ Values from the correlation between the quantity of plasmid and the number of copies in holm oak, as described in Material and Methods.

**Table 3 plants-11-00304-t003:** Regeneration performance with only root development and conversion into plantlets (simultaneous root and shoot development) of different holm oak transformed and untransformed (WT) lines after 8 weeks on germination medium.

Line	Only Root		Conversion (Shoot + Root)			
	(%)	RL (mm)	(%)	RL (mm)	SL (mm)	NL
Lines Q8						
Q8-WT	58.0 ± 6.8	52.5 ± 5.9	38.9 ± 5.1	54.9 ± 7.5	10.8 ± 0.9	7.6 ± 0.8
Q8-Gin1	33.3 ± 14.7	88.8 ± 9.9	52.8 ± 16.4	69.7 ± 1.8	11.6 ± 1.2	7.9 ± 0.2
Q8-Gin2	11.1 ± 3.2	105.0 ± 18.4	69.5 ± 10.0	78.3 ± 7.4	12.1 ± 0.9	9.5 ± 0.7
Q8-Gin3	47.2 ± 12.1	58.9 ± 7.4	30.6 ± 4.7	42.2 ± 8.5	10.1 ± 1.9	9.5 ± 2.1
ANOVA I	ns	ns	ns	*p* < 0.05	ns	ns
Lines E2						
E2-WT	55.6 ± 10.1	48.5 ± 5.6	38.9 ± 10.1	71.1 ± 13.1	16.7 ± 1.8	8.0 ± 1.3
E2-Gin1	8.3 ± 3.4	100.0 ± 30.1	22.2 ± 8.5	62.3 ± 9.2	15.5 ± 0.4	4.8 ± 0.8
ANOVA I	*p* < 0.05	ns	ns	ns	ns	ns

Each value represents the mean ± standard error of 6 replications with 6 explants in each replicate. RL, root length; SL, shoot length; NL, leaf number. Data were analyzed using ANOVA I (*p* ≤ 0.05). ns, not significant.

## Data Availability

Not applicable.

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
