# Peer review of "Genetic Transformation of Quercus ilex Somatic Embryos with a Gnk2-like Protein That Reveals a Putative Anti-Oomycete Action"

_plants, 2022, doi:10.3390/plants11030304_

Round 1
Reviewer 1 Report
I found the manuscript well written and presented. The introduction was clear and precise, leading well to the presented story. Experiments are described in detail and were performed according to current standards, the results were presented in an organized way allowing the reader to clearly follow the line of experiments presented. In fact, I did not find anything of significance to criticize in the manuscript.
Author Response
Thank you very much for your kind evaluation.
Reviewer 2 Report
Being a keystone species, the decline of holm oaks population by oomycete P. cinnamomi risks an ecological disaster. Selective crossing of the plant species consumes too much time to produce resistant individuals. Therefore, the authors generated P. cinnamomi-resistant plant of holm oaks through genetic transformation of holm oaks embryonic lines with Gnk2-like gene. The successful production of transformants was evidenced by selection antibiotic resistance, GFP expression, and PCR (determining transgenic gene copy numbers). Then resistance to P. cinnamomi was confirmed by qPCR (determining the transcript level)and longer survival duration upon oomycete inoculation. Overall, the scientific story itself is clear even though there are some blemishes in English writing. The research experiments themselves have taken great effort, especially since holm oak is known to be recalcitrant toward genetic modification and takes so much time to grow. Some data polishing and improvement in terms of data presentation might be necessary.
Major concerns:
Line 122: while it is mentioned in the materials and method section (which is located around the end of the manuscript), the 4 embryogenic lines appear suddenly in Table 1 and in result section 2.1., without being introduced in the introduction section. That might reduce the ease of understanding. Explanation on which part(s) of the DNA codes each cell line genetically differs should also be described, if possible, rather than just mentioning the source of the embryogenic cell lines because table 1 is included as one main figure (i.e. part of major findings) in your manuscript. Also, perhaps it might be useful (for laypeople like me) to know, do these cells lines have distinguishable phenotypes from one another when they are grown into whole adult plants?
Line 126: why selection efficiency of Q8 is 44% and not (2.5/5.0*100%) -> 50%?
Line 162 [Figure 2]/ Line 178 [Table 2]/ Line 191 [Figure 3]: Inconsistency of number of transformed Q8 lines among them. Figure 2 and Table 2 displayed only 2, while in Figure 3, Q8 has 3 lines. Any reason for that?
Line 178[Table 2]: Why was the Ct value detected in Q8-WT and E2-WT? Could it be that a non-specific fragment was amplified? At least, the authors should describe the sequences of the amplified fragments.Line 240: [Figure 4] perhaps this figure can be improved: first, the initial conditions do not seem to be equal enough -> non-transformed plant (A) is bigger & stouter than the transformed plant (B) -> they don’t seem to be in the same age.
If the Cast Gnk2-like protein work to inhibit P. cinnamomi growth and infection as hypothesized by Santos et al., (2017), growth of its hyphae would be affected in the transgenic plants. The observation of the hyphae could provide clear cut evidence of the P. cinnamomi resitance in the transgenic plants.
Also, it might reinforce your conclusion if you can show survival rates in ‘control condition’ -> plants grown in the same “in vitro tolerance assay” medium but without the oomycetes, to see if growing the plants in the medium itself trigger necrosis or not…
Line 312: Please provide evidence (e.g. image) to this “The in vitro behavior and growth of these transgenic plants were not affected by overexpression of the Cast_Gnk2-like gene” (more than what is implied in Table 3) and perhaps put it as one main figure -> It is actually rather important data since the value of pest-resistant plant variants will be reduced by if it has slow growth phenotype.
Minor concerns:
Line 29: “… survive more days …” -> “… survive longer …”
Line 41: “Oak decline has caused consists of the death of thousands of cork oaks and holm oaks…” ?
Line 77: “… after phenotyping with P. cinnamomi through root inoculation” -> “… after phenotyping through root inoculation with P. cinnamomi”
Line 82: what do you mean by” noninoculation conditions”? Do you mean ‘sterile condition’, ‘devoid of oomycete’?
Line 107: the first mention of kanamycin -> “… of kanamycin (kan)-resistant explants …”
*“Line 109”: “… being the line Q8 the one with the highest percentage (5%)… “ -> “… with Q8 being the highest percentage (5%).”
Line 115: “… but supplemented with 150 mg/ L kanamycin rather than 50 mg/ L initially…”
Line 120: “Selection efficiency was ranged from 44.4 to …”
*Line 121: “highest figures for selection efficiency (75%) were was found in E2 genotype”
Line 122: “Effect of EHA105pKWG2D-Gnk2-transformed holm oak somatic embryo genotypes on percentage of … in selection medium of holm oak somatic embryos transformed with the strain EHA105pKWG2D-Gnk2.”
Line 135: “Bar, 1 mm” -> “Bar: 1 mm”
Line 140: putatively
*Line 151: “… expression of the gene in the generated transgenic lines of holm oak generated.”
Line 162: [Figure 2] the lane legends ‘M P 1 2 3 4 5’ positioning doesn’t seem to be perfectly aligned with the corresponding PCR amplification result. Also, why suddenly only 2 Q8 transformed lines are checked? I thought the authors isolated 3 from Q8 (Line 143)?
Line 204: “… only root development and plant recovery were obtained from somatic embryos of 4 holm oak transgenic lines and in their corresponding wild-type lines.” -> “… root-only development and whole plant recovery were observed from in somatic embryos of 4 holm oak transgenic lines and in their corresponding wild-type lines.”
Line 205: “Among lines derived from Q8, conversion rates were ranged from … although and without significant differences to one another.”
Line 206: “… number of days survived survival days of by the transgenic and non-transformed lines was evaluated …”
Reviewer 3 Report
Thank you for giving me the opportunity to evaluate this manuscript which I found excellent and reflecting a soundly-based piece of research.
This manuscript is publishable pending a minor revision.
The text is written in a clear and proper English in terms of grammar and typos. I did find a few minor problems but since they do not interfere with comprehension of the text I decided no to include an annotated file. The authors may find easily instances where a minor English revision is required. These include the use of the verb to be before ranged (was/were ranged) which is of course incorrect or a few double uses of articles.
In fact, I only have three issues of concern:
1) Table 1: I found it rather odd that no significant difference was found in the conversion rate percentage between Q8-Gin 2 (69.5%) and Q8-Gin3 (30.6%). Please double check ... may be ANOVA is not the best test to be used here. Did you verify the normality of data distribution ? after all you had 6x6 replicates ...
2) L. 255-256 : I would avoid the use of the term resistance and use tolerance instead because the transformed plants survived longer after infection but still ultimately died also.
3) You have completely overlooked mentioning the use of Southerns to determine gene copy number and yet they are the ultimate and most precise tool to do so. Perhaps some mention to it should be made while underlining that you chose to use qPCR instead and taking it from there to point out the use of this as an alternative.
Round 2
Reviewer 2 Report
The manuscript looks much improved. I have found only a few points of concern.
Line 107, Table 1: I still cannot understand how the selection efficiency of Q8 is calculated. According to the table that the author attached, %selection efficiency could be 50%, right?
Online resource 4: The photo of E2-WT should be replaced. It looks blueish.
Author Response
See File.
